# Integrating WGCNA, TCN, and Alternative Splicing to Map Early Caste Programs in Day-2 Honeybee Larvae

**DOI:** 10.3390/genes16121409

**Published:** 2025-11-26

**Authors:** Xiang Ding, Jinyou Li, Dan Yue, Runlang Su

**Affiliations:** 1University of Chinese Academy of Sciences, Beijing 100049, China; dingxiang18@mails.ucas.ac.cn; 2Faculty of Computing & Data Sciences, Boston University, Boston, MA 02215, USA; yukili@bu.edu; 3College of Animal Science, Yunnan Agricultural University, Kunming 650201, China; yuedan2015@126.com

**Keywords:** *Apis mellifera*, WGCNA, alternative splicing, deep learning, caste

## Abstract

Background/Objectives: The larval stage plays a pivotal role in determining caste and sex in *Apis mellifera*. This study integrates RNA-seq, WGCNA, and alternative splicing analyses to explore gene expression differences among 2-day-old worker, drone, and queen larvae. Methods: RNA-seq was conducted on 2-day-old larvae from all three castes. Differential expression, WGCNA, and alternative splicing patterns were investigated. A deep learning TCN model was trained using WGCNA-derived modules and demonstrated high classification accuracy. Results: The TCN model highlighted a top-10 gene set, including *PDHB*, *Fibroin3*, and *LOC724161*. Significant caste- and sex-specific splicing events were detected in *Tk*, *Csd*, and *Fem*, with AF events being most prevalent. Splicing differences between sexes exceeded those observed among castes. Conclusions: The 2-day-old larval stage is crucial for both caste and sex differentiation in honeybees. This study identifies key genes and splicing events, offering new insights into the molecular mechanisms underlying caste formation and sex determination.

## 1. Introduction

Honeybees are highly social insects distinguished by a complex division of labor, with workers, queens, and males fulfilling distinct biological roles within the colony [1,2]. Although workers and queens are both Female, their caste fate is primarily determined by larval nutrition. Queens develop functional ovaries and are responsible for egg laying, whereas workers are sterile and perform tasks such as brood care, foraging, and nest construction. Males (drones) appear seasonally and mate with queens to sustain colony reproduction [3]. These caste differences are reflected at behavioral, physiological, and molecular levels.

Honeybee development progresses through the embryonic, larval, pupal, and adult stages. The larval stage is particularly dynamic, characterized by rapid growth and significant transcriptional and epigenetic shifts that shape caste differentiation. During the first three days, all larvae consume royal jelly, allowing Female larvae under three days to retain dual developmental potential [4]. The 2-day larval stage is considered a critical window for caste specification in both honeybees and bumblebees [5]. Although chromosomal conformation shows minimal differences between 2-day-old queen and worker larvae, chromatin architecture remains closely tied to transcriptional regulation [6]. DNA methylation also plays a central role, with queens generally showing lower methylation levels than workers [7]. At later larval stages, queens show widespread demethylation and a substantially greater number of differentially expressed genes than at the 2-day stage [8], underscoring the early larval period as a key period for regulatory divergence.

Previous transcriptomic studies have primarily focused on queen–worker comparisons, while information on male larvae remains limited. Thousands of genes differ between queen and worker larvae, influencing metabolism, development, and physiology. For example, antioxidant genes (*MnSOD*, CuZnSOD, *catalase*, *Gst1*) are associated with oxidative stress regulation [9], and mitochondrial translation factors such as AmIF-2mt and cytochrome c are linked to queen developmental rate [10]. Caste development consists of an initial dual-potential phase followed by irreversible commitment to a specific developmental trajectory [11]. In males, sex determination is governed by the complementary sex determination (*Csd*) gene, which regulates the downstream Feminizer (Fem) gene. Hemizygous *Csd* alleles trigger Female-pattern Fem splicing to produce males, while heterozygosity induces Female development [12,13]. Many sex-specific transcriptional differences arise from alternative splicing (AS) [14], and CRISPR/Cas9 knockdown of Fem has demonstrated its role in regulating the splicing of multiple downstream genes involved in sexual differentiation [15].

Alternative splicing is a key post-transcriptional mechanism that greatly expands proteome complexity. Approximately 95% of human genes undergo AS [16], and AS contributes to development and diverse cellular processes by generating multiple mRNA isoforms from a single gene [17]. Splicing regulation is essential across species [18]. In honeybees, AS is associated with epigenetic modifications, such as DNA and m6A methylation, that shape caste-specific transcript variation [7,19,20].

To elucidate transcriptional and splicing differences across castes, this study analyzes transcriptomes of 2-day-old worker, queen, and male larvae of the western honeybee using WGCNA and alternative splicing approaches. This study aims to identify differentially expressed genes, characterize sex-related transcriptional signatures, and investigate how AS contributes to caste-specific developmental pathways.

## 2. Materials and Methods

### 2.1. Sample Collection

Transcriptome data from 2-day-old Italian honeybee larvae were retrieved from the NCBI database, with six biological replicates for each caste (workers: SRS1249139; queens: SRS1263242; drones: SRS1263244) [21]. To validate the RNA-seq findings, we also collected an independent qPCR cohort consisting of 2-day-old *Apis mellifera* worker, queen, and drone larvae, each group including six biological replicates, under matched environmental and experimental conditions.

For sex-level analyses, workers and queens, both genetically Female (diploid), were combined into a non-drone (ND) group to increase Female sample size and better capture sex-related expression patterns shared across Female larvae. In comparison, caste-level comparisons (queen vs. worker) were performed without pooling to preserve caste-specific transcriptional features and avoid confounding effects between sex and caste.

Larvae were sampled from *Apis mellifera* colonies maintained near Yiwu Industrial and Commercial College. Following collection, samples were immediately processed for RNA extraction and qPCR under controlled laboratory conditions. Rearing conditions were maintained at approximately 34 °C and 70% relative humidity, with no additional environmental enrichment. All sample processing and qPCR assays were carried out by Anhui Gaohe Biotechnology Co., Ltd. (Anqing, China).

### 2.2. Mapping and Quality Control of RNA-Seq Data

Quality control of the clean sequencing data was performed using FastQC to generate base composition and quality distribution plots. Raw reads were processed with Fastp (v0.23.2) for adapter removal and quality trimming using the following parameters: automatic adapter detection for paired-end reads, Q20 base trimming, filtering of reads with >40% low-quality bases or >5 ambiguous bases, and retention of reads ≥50 bp. Clean reads were aligned to the *Apis mellifera* reference genome (Amel_HAv3.1) using Hisat2 (v2.2.1) [22]. By default, Hisat2 uses a 20 bp seed length, permits soft clipping, applies mismatch penalties of 6 and 2, and reports up to 5 valid alignments per read. Alignment outputs in SAM format were converted to BAM files, sorted with Samtools (v1.6) [23], and quantified with FeatureCounts [24] to obtain gene-level read counts. Differential expression analysis was performed using the DESeq2 algorithm integrated within Trinity (v2.15.1) [25,26]. *p*-values were adjusted using the Benjamini–Hochberg false discovery rate (FDR), and genes with FDR < 0.05 and |log_2_FC| > 1 were considered significantly differentially expressed. To statistically compare the relative contributions of sex and caste to transcriptional variation, a two-way ANOVA was performed for each gene with sex (male vs. female) and caste (worker vs. queen vs. drone) as factors. Effect sizes were quantified using partial eta-squared (ηp^2^), and the significance of main effects was assessed at α = 0.05.

### 2.3. PCA

To assess sample consistency, principal component analysis (PCA) was conducted on the gene expression matrix. Before performing PCA, the data were standardized using the scale function in R to ensure comparability across all dimensions.

### 2.4. GO Enrichment Analysis of Differentially Expressed and Alternatively Spliced Genes

The longest CDS-derived protein sequences from the *Apis mellifera* reference genome (*Amel_HAv3.1*) were annotated for Gene Ontology (GO) using emapper.py (v2.1.10) [27] with the EggNOG database [28]. The resulting annotation file was converted into org.db format, and GO enrichment analyses were performed using the clusterProfiler package (v4.4) in R. For both GO and KEGG enrichment analyses, multiple testing correction was applied using the Benjamini–Hochberg FDR method, and only terms with FDR < 0.05 (*p*-value < 0.05 and adjusted *p*-value < 0.05) were retained.

### 2.5. Alternative Splicing Analysis

Alternative splicing analysis was conducted using SUPPA (v2.4) with the reference genome annotation (GTF) [29,30]. Major SUPPA splicing event types, including skipped exon (SE), mutually exclusive exon (MXE), alternative 5′ splice site (A5), alternative 3′ splice site (A3), and retained intron (RI), were quantified. Transcript abundance was estimated using Salmon (v1.10.1), and ΔPSI values were calculated for all events. From individual biological replicates (*n* = 6 per phenotype group: worker, drone, queen). Percent-spliced-in (PSI) values were calculated for each splicing event in each replicate, and group-level ΔPSI values were derived by comparing mean PSI values between phenotype pairs (worker vs. queen, worker vs. drone, drone vs. queen). Differential splicing was identified using a ΔPSI threshold of ≥0.1, together with statistical significance (*p* < 0.05) and Benjamini–Hochberg FDR correction. Only events meeting both ΔPSI and FDR criteria were considered significant.

### 2.6. Weighted Gene Co-Expression Network Analysis (WGCNA) and TCN Modeling

After removing outlier samples through pairwise correlation analysis and hierarchical clustering, weighted co-expression networks were constructed using the WGCNA R package(v1.73) (https://cran.r-project.org/web/packages/WGCNA/index.html) (accessed on 23 November 2025) (networkType = “signed”, mergeCutHeight = 0.25, minModuleSize = 30, deepSplit = 2). Automatic blockwise module detection was applied, using a data-driven soft threshold to approximate a scale-free topology. Module eigengenes (MEs) were calculated and correlated with experimental conditions to identify trait-associated modules. Gene–gene regulatory networks from selected modules were exported and visualized in Cytoscape (v3.10.1).

To refine biomarkers from trait-associated modules and develop a phenotype classifier, a Temporal Convolutional Network (TCN) was trained using the module gene expression matrices (https://github.com/surunlang-creator/Drone_torch) (accessed on 23 November 2025). Expression values were log-transformed and z-scored for each gene, and a fixed random seed (5678) ensured reproducibility. Data were partitioned into training, validation, and test sets at a 6:2:2 ratio with phenotype stratification (worker, drone, queen) to maintain balanced class representation across splits. The TCN architecture consisted of three dilated causal convolutional layers with kernel size 3, dilation rates of 1, 2, and 4 to capture multi-scale temporal dependencies, and channel dimensions of 64, 64, and 128, respectively. Each convolutional layer was followed by batch normalization and dropout (rate = 0.3) for regularization, with final classification performed through adaptive average pooling and a fully connected layer. The model was implemented in PyTorch (2.9.1), optimized using the Adam optimizer (learning rate = 0.001, weight decay = 1 × 10^−4^) with ReduceLROnPlateau scheduler (factor = 0.5, patience = 10), and trained for up to 100 epochs with early stopping (patience = 20) based on validation loss to prevent overfitting. Model performance was assessed on the held-out test set using accuracy metrics, precision, recall, F1-score, training/validation loss curves, and a confusion matrix. Gene-level importance scores were generated from the trained model by calculating composite scores integrating prediction accuracy, expression variance (R^2^ values), and pattern-specific weights, with the top 10 features per phenotype retained for downstream interpretation and network visualization. These candidate genes were subsequently used to guide qPCR validation experiments.

## 3. Results

### 3.1. Overall RNA-Seq Results and Quality Control

Illumina RNA sequencing produced an average of 3.42 Gbp of raw reads per sample. After quality filtering, approximately 3.38 Gbp of clean reads per sample were retained for downstream transcriptome analysis. These clean reads demonstrated high sequencing quality, with average Q20 and Q30 values both reaching 100%, and an average GC content of 38.00% (Appendix A). Alignment to the reference genome with HISAT2 showed that more than 88% of transcripts were uniquely mapped, as determined by sliding-window density analysis. Across all 18 samples, 645 million clean reads were uniquely aligned, yielding an average mapping rate of 93% (Appendix A).

### 3.2. Hierarchical Clustering and Principal Component Analysis of Samples

Hierarchical clustering and PCA (Figure 1) demonstrated strong within-group consistency and clear separation among queen, worker, and drone samples. PCA was performed on all expressed genes following standard scaling, with PC1 and PC2 accounting for the majority of the variance. Outlier detection based on sample-to-sample distance matrices and projections onto the first two principal components revealed no abnormal samples. The tight clustering of biological replicates within each caste further indicated the absence of batch effects. These findings establish a reliable basis for subsequent differential expression and alternative splicing analyses.

### 3.3. Analysis of Differentially Expressed Genes (DEGs)

To investigate gene expression differences among workers, queens, and drones, a differential expression analysis was performed, and DEGs across three pairwise comparisons, drones vs. queens, workers vs. drones, and workers vs. queens (Figure 2A), were compared. The smallest expression difference was observed between workers and queens, with 247 DEGs identified (DESeq2, adjusted *p* < 0.05, |log_2_FC| > 1), whereas drones and queens showed the largest divergence, with 782 DEGs detected. This difference was statistically significant (x^2^ test, *p* < 0.001), indicating that sex-based differences exceed caste-based differences by 3.2-fold. Volcano plot analysis further highlighted several genes demonstrating substantial fold changes (Figure 2B). Genes with high fold change between drones and queens included *LOC551527*, *LOC410733*, *LOC411233*, and *LOC411188*. Genes strongly differentiated between workers and drones included *CPR5*, *LOC100578919*, *LOC411884*, *LOC726134*, and Fem, while *LOC724536*, *LOC100578046*, and vg showed prominent differences between workers and queens. These highly variable genes are likely to play important roles in honeybee sex differentiation, physiological traits, and behavioral regulation.

### 3.4. Gene Expression GO Analysis

To explore the functional roles of the differentially expressed genes among the three castes, the top 11 most highly expressed genes were selected for GO enrichment analysis (Figure 3). The DEGs between drones and queens were predominantly enriched in biological processes related to development and growth, sex differentiation, male sex differentiation, primary sexual characteristic development, and male sex determination. DEGs identified between workers and drones were enriched in pathways associated with sex differentiation, male sex differentiation, primary male sexual characteristic development, male anatomical structure morphogenesis, and primary sex determination. In comparison, DEGs between workers and queens were mainly enriched in processes linked to embryonic organ morphogenesis, brain development, and vitellogenesis.

### 3.5. WGCNA

To investigate genetic correlations and caste-specific expression patterns among workers, drones, and queens, WGCNA was performed on all aligned genes (9,939 in total). After sample clustering (Figure 4A), missing values were processed and outliers removed. To ensure that the network approximated a scale-free topology, a soft-threshold power of 24 was selected (scale-free R^2^ = 0.80) (Figure 4B,C). Using the one-step construction method, a co-expression matrix was generated, and 9 gene modules were identified via dynamic tree-cutting (Figure 4A,D). The turquoise module (1751 genes) and gray module (5412 genes) showed the strongest association with drones (correlation = 0.82, *p* = 3 × 10^−5^) (Figure 4D). The brown module (368 genes) displayed the highest correlation with queens (correlation = 0.82, *p* = 3 × 10^−5^), while the yellow module (210 genes) was most strongly associated with workers (correlation = 0.70, *p* = 0.001). These module–trait relationships highlight significant specificity in gene expression across castes, suggesting that unique transcriptional programs underpin the distinct biological functions and behaviors of each group.

A TCN deep learning model was applied to further evaluate the phenotype-associated modules identified by WGCNA. The dataset was divided into training, validation, and test sets at a 9:1:1 ratio. As shown in Figure 5A,B, both training and validation loss rapidly converged toward zero without evidence of overfitting, and accuracy for both sets reached nearly 100% within the first 30 epochs. The model achieved perfect classification accuracy (100%) on the independent test set, demonstrating strong generalization performance. Using the trained TCN model, top-ranked representative genes were extracted for the worker, drone, and queen phenotypes. These indicator genes were visualized using neural network diagrams (Appendix A), and their expression differences among castes were statistically significant (Appendix A). qPCR validation (Figure 6; Appendix A; Supplemental Appendix A) further confirmed the accuracy and reliability of the TCN-derived gene candidates.

### 3.6. Statistical Analysis of Differential Genes and Events in Alternative Splicing Events

To investigate differences in gene isoform expression among workers, drones, and queens, a detailed analysis of alternative splicing patterns was performed. Consistent with the differential expression results, workers and queens showed the smallest divergence, with the fewest alternatively spliced genes and corresponding transcripts, 25 and 20, respectively (Figure 7A). In comparison, drones and workers showed the greatest differences, with 51 alternatively spliced genes and 46 associated transcripts (Figure 7A). Among all splicing categories, AF (Alternative First Exon) events were the most prevalent across castes, totaling 27 in drones, 20 in queens, and 15 in workers (Figure 7B). These patterns likely reflect caste-specific regulatory requirements shaped by distinct physiological roles. In particular, the higher number of alternatively spliced genes and transcripts, especially the increased AF events, between drones and workers may indicate the need for more complex regulatory mechanisms governing drone reproduction, reproductive organ development, and mating-associated behaviors.

### 3.7. GO Enrichment Analysis of Alternative Splicing

GO enrichment analysis was performed to explore the functional implications of alternative splicing events among the three honeybee castes. The results revealed both shared and caste-specific patterns related to developmental and functional differentiation. Drones and queens displayed strong similarities in reproductive and developmental regulatory pathways, particularly those associated with reproductive organ growth and regulation. Drones and workers showed overlap in processes linked to developmental maturation and core reproductive functions, reflecting common features of sexual differentiation. Queens and workers were closely aligned in pathways related to structural formation and functional specialization, consistent with their roles in morphogenesis and social division of labor (Figure 8). These results provide important molecular insights into the mechanisms driving social differentiation and functional regulation in honeybees.

### 3.8. Alternative Splicing of Genes Related to Sex Determination

To investigate how alternative splicing contributes to sex determination and differentiation among the three honeybee castes, three genes—two known sex-determination genes (*Csd* and *Fem*) and an additional gene, *Tk* (Figure 9A,B)—were analyzed. All three genes demonstrated alternative splicing events across castes. The *Csd* gene (chr3:11,135,000–11,146,004) showed a unique intron retention event in drones and an exclusive exon-skipping event in queens. Furthermore, at the genomic region chr3:11,781,139–11,780,712, queens lacked an intron-retention event present in both workers and drones. For *Fem*, drones displayed the greatest number of exon-skipping and intron-retention events.

In comparison, workers and queens shared similar splicing patterns, with workers demonstrating one additional exon-skipping event. In the *Tk* gene (chr7:3,735,230–3,735,382 and chr7:3,735,570–3,735,914), queens lacked two intron retention events observed in workers. These results highlight caste-specific splicing patterns among genes with large fold changes and suggest their involvement in honeybee developmental and caste-differentiation processes.

To validate these predicted splicing events, primers targeting specific transcript regions were designed, and qPCR analysis (Figure 10; Appendix A) was performed. The qPCR results confirmed the predicted alternative splicing patterns, showing complete consistency with the caste-specific events identified for all three genes.

## 4. Discussion

To identify genes with substantial fold changes and caste-specific alternative splicing patterns in 2-day-old western honeybee workers, drones, and queens, transcriptomic datasets from the NCBI database were first retrieved. More stringent data-processing standards were applied than those used in previous studies. This approach yielded a larger set of differentially expressed genes compared with previous reports [21] (Figure 2A). Further, PCA revealed a clear intrinsic structure among samples, with strong within-group consistency for workers, queens, and drones. These results confirm both the robustness of the data and the reliability of the filtering strategies used in this study.

In the analysis of differentially expressed genes (Figure 2A), more stringent criteria were applied to the raw data, resulting in a substantial increase in the number of detected DEGs compared with the findings of He et al. The number of DEGs between males and queens increased by 64%, while the numbers for workers vs. males and workers vs. queens nearly doubled. Among the three castes, males and queens demonstrated the highest number of DEGs, whereas workers and queens showed the fewest. In comparison to the observations by Christina Vleurinck et al. [31], reporting that caste-related DEGs during the pupal stage were shared mainly between sexes, these results demonstrate that at the 2-day larval stage, sex-related differences significantly exceed caste-related differences (two-way ANOVA across all expressed genes: mean sex effect ηp^2^ = 0.58 ± 0.12, mean caste effect ηp^2^ = 0.31 ± 0.09; paired *t*-test, *p* < 0.001). Specifically, the drone vs. queen comparison yielded 782 DEGs, significantly more than the 247 DEGs between workers and queens (Fisher’s exact test, *p* = 3.4 × 10^−8^, odds ratio = 3.17). In the volcano plots (Figure 2B), several high-fold change genes associated with honeybee growth, development, and sex determination were identified. Among them, *vg* encodes vitellogenin, a major Female-specific yolk glycoprotein precursor. As a Female-specific glycolipoprotein, vitellogenin supports fat body development and enhances immunity and stress resistance in honeybees [32,33]. Another essential gene is *Fem*, a key regulator of sex determination [34]. Its expression is controlled by the complementary sex determiner (*Csd*) gene, and only Female-specific splicing of *Fem* produces functional *Fem* protein required for Female development [34]. *CPR5* encodes an epidermal protein that interacts with chitin and contributes to exoskeleton formation and stability, playing important roles in structural integrity, water retention, and environmental protection [35]. *LOC411233* encodes MPP10, a core component of the U3 small nucleolar ribonucleoprotein complex involved in ribosomal RNA processing and ribosome assembly [36]. Since ribosomes are essential for protein synthesis, this gene is critical for normal cell growth and development. Several highly expressed genes with previously unknown functions, such as *LOC724536* and *LOC725439*, were also identified that may contribute to developmental processes in honeybees.

GO enrichment of DEGs among workers, drones, and queens (Figure 3) showed that most of the differences between drones and queens, and between drones and workers, involved sex differentiation, indicating pronounced sex-specific divergence during the larval stage. In comparison, DEGs between workers and queens were primarily enriched in pathways related to brain development and organogenesis. These differences likely reflect nutritional divergence: the protein and 10-HDA content in the diet of 2-day-old queen larvae is significantly higher than that of worker larvae [37], driving caste-specific developmental outcomes. This nutritional influence complies with the worker–queen DEG patterns observed in the volcano plot (Figure 2B). These findings indicate that both sex differentiation and caste-specific gene expression differences are already well established by the 2-day larval stage in workers, drones, and queens.

Integrating WGCNA with deep learning enabled us to substantially narrow the candidate gene pool while maintaining strong phenotype-discriminative power, a strategy that has proven effective in various disease-gene discovery studies [38,39,40,41]. Using phenotype-associated modules derived from WGCNA, the TCN classifier identified a concise set of high-priority genes. Investigation of the top 10 weighted features revealed several biologically meaningful signals. *PDHB* showed the most striking differential expression pattern, with expression levels in non-queen larvae 6.87-fold higher than in queen larvae (log_2_FC = 2.78, adjusted *p*-value = 1.2 × 10^−5^, 95% CI: 5.93–7.81-fold). *PDHB* encodes the β-subunit of the pyruvate dehydrogenase complex, a central metabolic enzyme that channels glycolytic carbon into the TCA cycle and interfaces with lipid and energy metabolism. Its reduced expression in 2-day-old queen larvae aligns with previous findings that queens and workers adopt distinct metabolic strategies early in development [42]. Studies in other biological systems also link *PDH/PDHB* activity to developmental processes, including skeletal and tissue growth, suggesting that early *PDHB* modulation may couple nutrient signaling to downstream morphogenetic pathways [42]. Although direct causal relationships in honeybees remain unconfirmed, these results demonstrate that caste-specific metabolic–developmental divergence is already established at the larval stage.

Caste-biased expression of *Fibroin3*, a member of the fibroin family that forms the primary structural component of the insect silk gland and contributes to the architecture of the cocoon and nest, was also identified. In honeybees, larval silk plays an essential role in reinforcing comb stability; therefore, early divergence in *Fibroin3* expression provides molecular support for caste-specific differences in structural and behavioral potential related to nest building [43,44]. This interpretation is in good agreement with qPCR results, which showed elevated *Fibroin3* expression in the non-drone group. This pattern aligns with the adult division of labor, in which drones lack functional wax glands and do not participate in comb construction. In comparison, workers develop well-developed wax glands, which are essential for nest building. However, direct evidence linking fibroin genes to wax gland ontogeny remains limited. A more cautious interpretation is that *Fibroin3* marks early specification of structural and behavioral programs associated with nest architecture rather than directly influencing wax gland formation [45].

These results demonstrate how a WGCNA-to-deep learning framework can distill high-dimensional transcriptomic data into a compact, interpretable, and phenotype-predictive gene set, while highlighting mechanistic hypotheses linking metabolic regulation (*PDHB*) to caste-specific structural programs (*Fibroin3*). Future studies should integrate multi-cohort validation with functional perturbation approaches, such as RNAi, CRISPR, or nutrient and hormone manipulations, to test causality, and employ cell-type-resolved analyses to map these molecular signatures onto the developmental lineages that drive early caste divergence.

In the analysis of differential alternative splicing events (Figure 7A), the fewest selectively spliced genes and transcripts between workers and queens were observed. In comparison, males showed substantially more selectively spliced genes than either workers or queens. This pattern mirrors the differential gene expression results (Figure 2A), indicating that during the 2-day larval stage, sex-specific differences in alternative splicing are more pronounced than caste-specific differences. Alternative First Exon (AF) events were the most frequent among all splicing types (Figure 5B). Because 2-day-old honeybee larvae are undergoing rapid growth and developmental transitions, selective splicing of the first exon can generate protein isoforms with distinct functional domains or subcellular localizations, increasing protein diversity [46]. GO enrichment analysis of these alternative splicing events further revealed that the associated genes participate in a wide range of processes, including growth, development, and sexual reproduction. These findings suggest that alternative splicing contributes broadly to the regulation of sex differentiation and other physiological changes in honeybee larvae.

Because *Fem* showed substantial expression differences between workers and males in Figure 2B and Figure 9B, alternative splicing patterns in *Fem*, its upstream regulator *Csd*, and the behavior-associated gene *Tk* were further analyzed. *Csd* is a honeybee-specific sex-determination gene [44], and its RS domain and proline-rich region are known to mediate protein-binding interactions [45,46]. Variations in these amino acid sequences may therefore alter protein–protein interactions that are critical for splicing regulation. In this study, both *Csd* and *Fem* demonstrated significantly higher expression in males than in queens and workers (Figure 7), while *Tk* showed no substantial differences in overall expression. Consistent with this, the alternative splicing patterns of the three genes showed a similar trend (Figure 9C): *Tk* had relatively simple splicing variants across castes, whereas *Csd* and *Fem* displayed much more complex splicing profiles, particularly in males. These findings suggest a strong relationship between gene expression levels and the splicing complexity of their transcripts.

All three genes, *Csd*, *Fem*, and *Tk*, demonstrated caste-specific splicing variants, with males showing the most complex patterns. This supports the idea that alternative splicing influences not only caste development [11] but also sex differentiation. A mechanistic association between the alternative splicing variants of *Fem* and *Csd* is further hypothesized. Previous studies have shown that the binary switch in honeybee sex determination is controlled through selective splicing of *Fem* transcripts in response to *Csd* heterozygosity [45,46,47,48,49,50]. Similarly, in *Drosophila*, the *tra* gene, homologous to *Csd*, acts as a switch gene required for Female development through sex-specific splicing regulation [51,52]. Our analysis demonstrates that *Csd* and *Fem* undergo distinct alternative splicing patterns in workers, males, and queens, underscoring the utility of splicing-based approaches for studying sex determination in honeybees. These findings also provide new avenues for future research on developmental regulation and behavioral differentiation in honeybee castes.

In summary, the combined results from WGCNA and alternative splicing analyses demonstrate that 2-day-old worker, drone, and queen larvae are at a pivotal stage of sex differentiation, physiological transition, and organ development. Both the overall number of differentially expressed genes and the number of differentially spliced genes indicate that sex-related differences are more pronounced than caste-related differences at this early stage. Tissue- and organ-specific developmental pathways appear to be shaped by distinct patterns of selective splicing, which likely drive the divergent gene expression profiles observed among the three castes. These findings improve our understanding of the molecular mechanisms underlying early larval development in workers, drones, and queens, and provide valuable insights into how alternative splicing influences developmental trajectories in honeybee larvae.

## 5. Conclusions

In this study, transcriptome data from 2-day-old worker, drone, and queen larvae of the western honeybee were reanalyzed using enhanced filtering, WGCNA, and alternative splicing approaches. Both caste- and sex-specific differences in gene expression and splicing, including key regulatory genes such as *vg*, *Fem*, and *Csd,* were identified. These findings underscore the substantial transcriptional and post-transcriptional regulation occurring at this early larval stage and provide valuable insights into the molecular mechanisms governing honeybee caste differentiation and sex determination.

## Figures and Tables

**Figure 1 genes-16-01409-f001:**
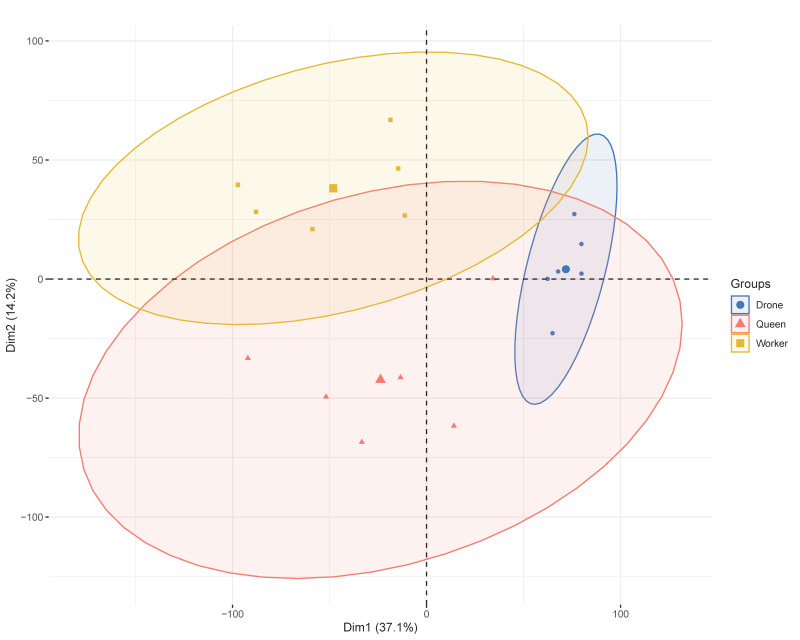
Principal component analysis (PCA) of the gene expression matrix for queens, workers, and drones (*n* = 6).

**Figure 2 genes-16-01409-f002:**
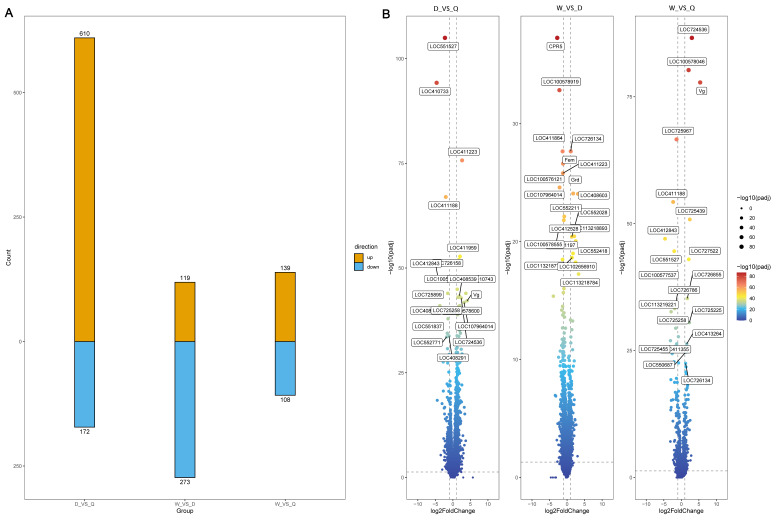
Differential gene expression analysis. (**A**) Statistical data of differentially expressed genes (D: Drone; W: Worker; Q: Queen). (**B**) Volcano Plots of Gene Expression in the Three Castes of Honeybee During 2-Day Larval Development (D: Drone; W: Worker; Q: Queen).

**Figure 3 genes-16-01409-f003:**
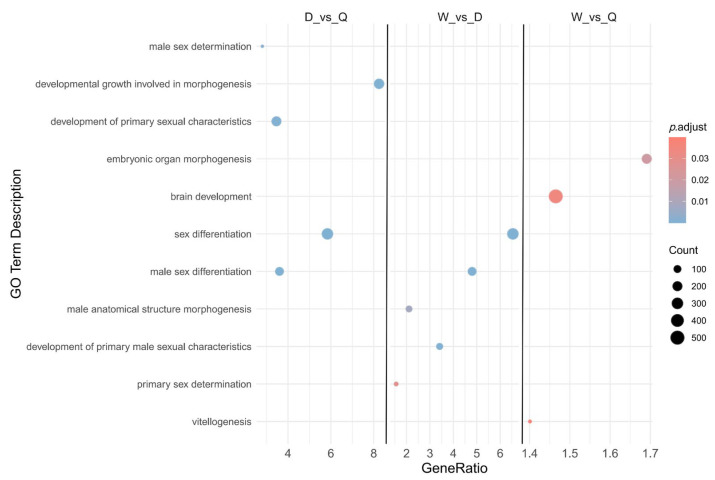
GO Analysis of Differentially Expressed Genes Among Three Castes of Honeybee (D: Drone; W: Worker; Q: Queen).

**Figure 4 genes-16-01409-f004:**
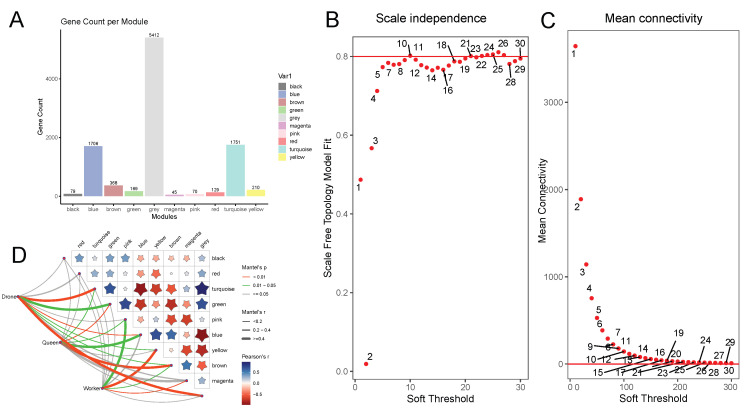
Construction of WGCNA Modules. (**A**) The distribution of gene numbers across the nine color modules. (**B**,**C**) Analysis of the scale–free topology fit index and average connectivity under different soft–thresholding powers. The red line indicates the soft–thresholding power of 24, where the correlation coefficient reaches 0.8. (**D**) Heatmap illustrating the correlation between gene expression and the three castes. The thickness of the lines indicates the strength of the correlation. Brown lines represent highly significant *p*-values, while green lines represent significant *p*-values. The stars highlight the correlations within each color module, with larger stars indicating stronger absolute correlation values.

**Figure 5 genes-16-01409-f005:**
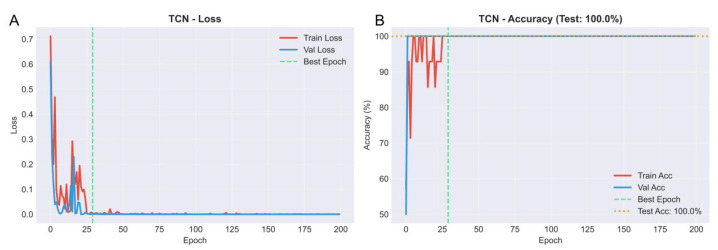
Training and validation performance of the TCN model. (**A**) Training and validation loss curves showing rapid convergence without overfitting. (**B**) Training and validation accuracy curves, with the model achieving 100% classification accuracy on the independent test set.

**Figure 6 genes-16-01409-f006:**
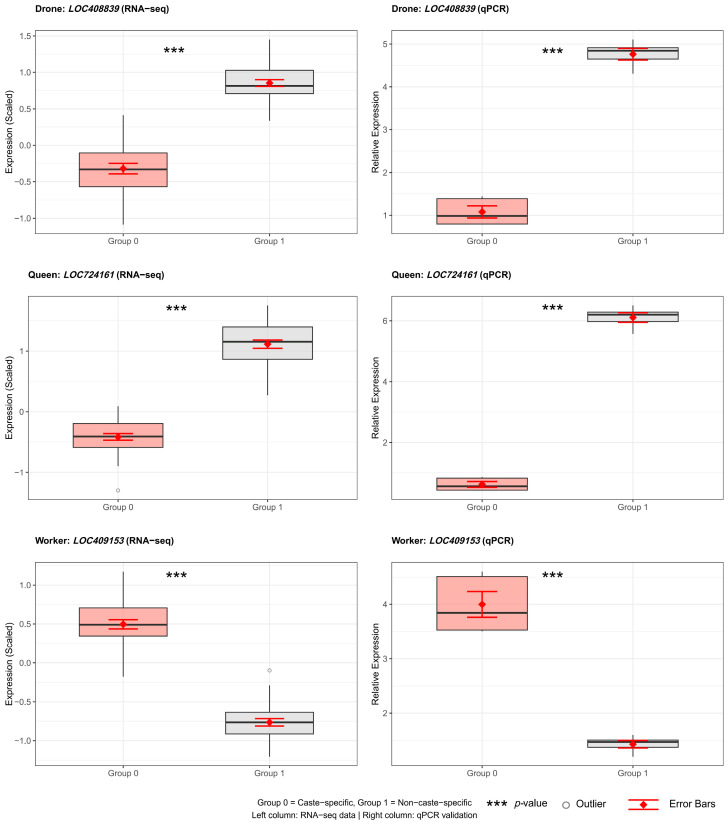
qPCR validation of RNA-seq data for caste-specific genes. Representative genes from drone (*LOC408839*), queen (*LOC724161*), and worker (*LOC409153*) showing concordant expression patterns between RNA-seq (left, *n* = 30/group) and qPCR (right, *n* = 5/group). Box plots show median, IQR, and mean ± SE (red diamonds with error bars). Group 0: caste-specific; Group 1: non-caste-specific. Statistical tests: Kruskal–Wallis (RNA-seq) and *t*-test (qPCR). *** *p* < 0.001 (Non-drone samples comprise pooled queens and workers at equal mass per replicate to test sex-level effects).

**Figure 7 genes-16-01409-f007:**
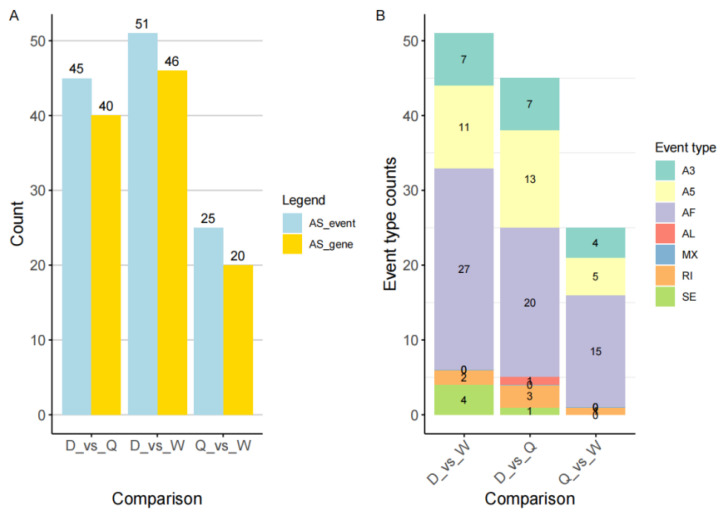
Alternative Splicing (AS) Events in the Three Castes of Honeybees. (**A**) The number of differentially spliced genes identified among the three honeybee castes (D: Drone; W: Worker; Q: Queen). (**B**) Types and distribution of alternative splicing events observed in the three castes of honeybees (D: Drone; W: Worker; Q: Queen). Alternative Splicing Events types: A3 (Alternative 3′ splice site, A3SS); A5 (Alternative 5′ splice site, A5SS); AF (Alternative First Exon, AF); AL (Alternative Last Exon, AL); MX (Mutually Exclusive Exons, MXE); RI (Retained Intron, RI); SE (Skipped Exon, SE).

**Figure 8 genes-16-01409-f008:**
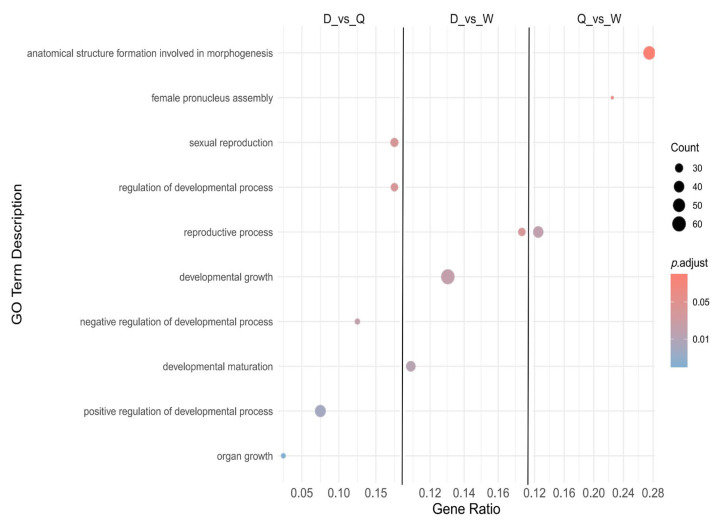
GO Enrichment Analysis of Alternative Splicing in the Three Castes of Honeybees.

**Figure 9 genes-16-01409-f009:**
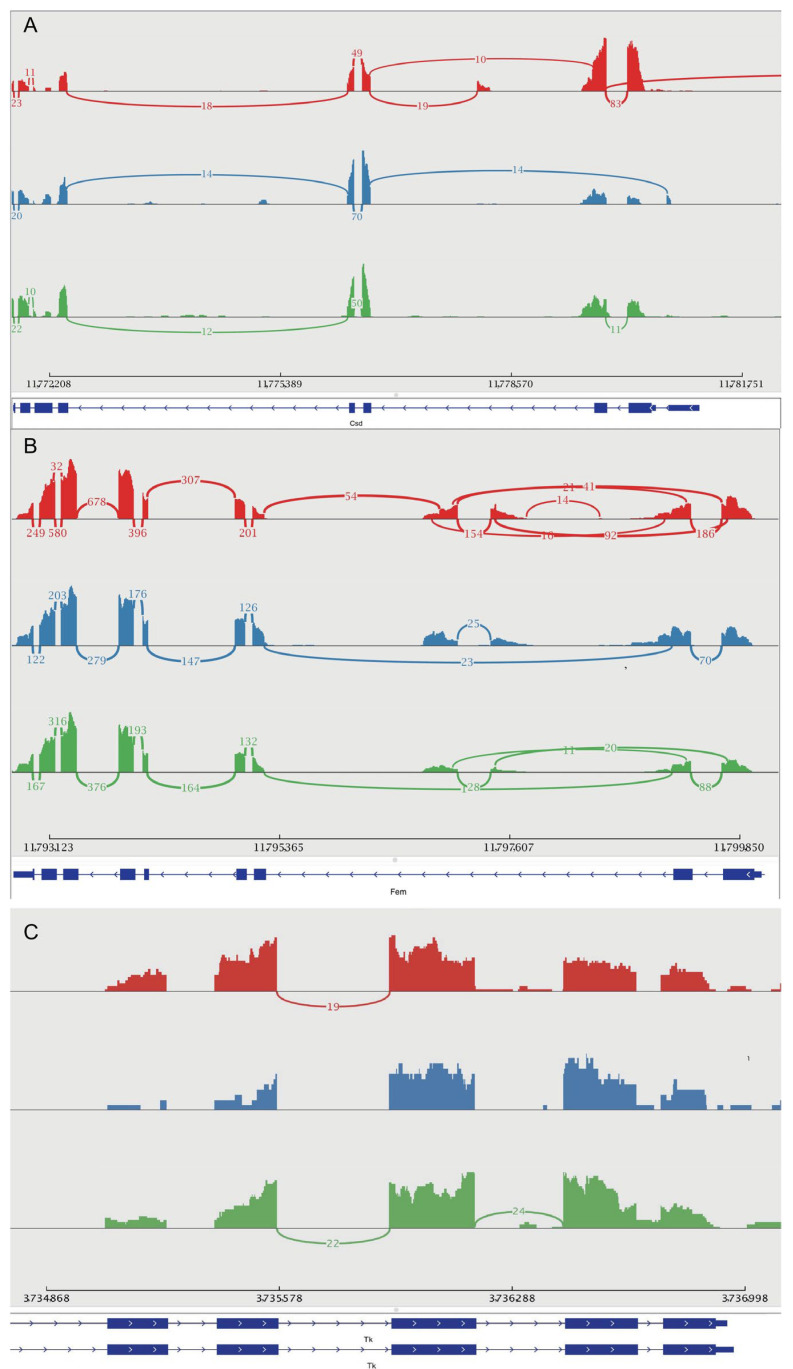
Sashimi Plots of Gene Splicing Events. (**A**) Sashimi plot showing the splicing patterns of the *Csd* gene. (**B**) Sashimi plot showing the splicing events of the *Fem* gene. (**C**) Sashimi plot showing the splicing patterns of the *Tk* gene. (Red: Drone; Blue: Queen; Green: Worker; Below each Sashimi plot are isoforms of the corresponding splicing event.The number on each arc indicates the number of reads that support the corresponding splicing junction.).

**Figure 10 genes-16-01409-f010:**
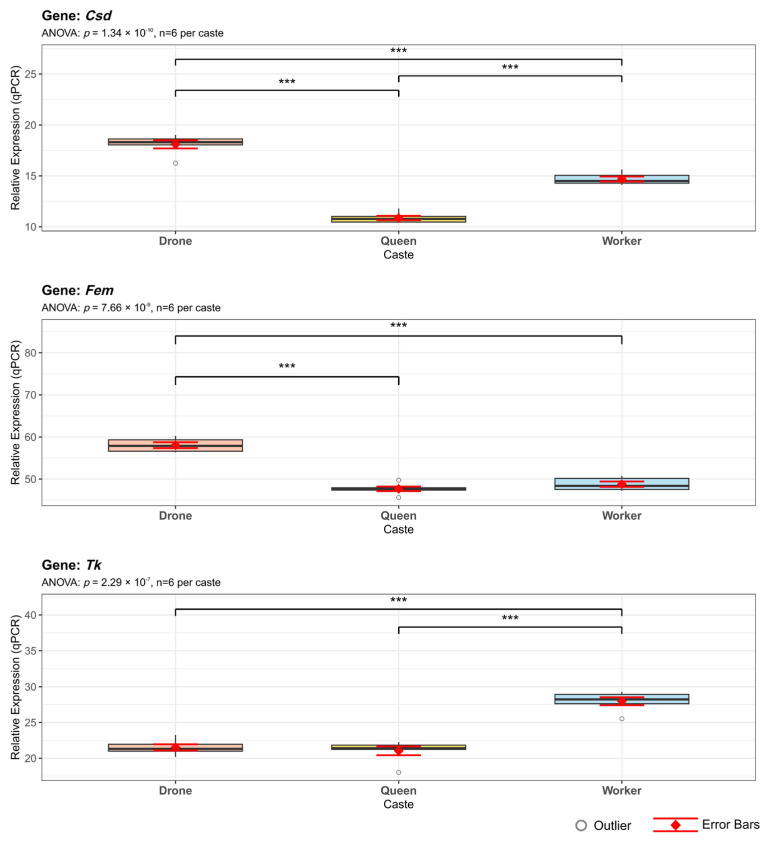
qPCR Validation of Caste-Specific Alternative Splicing in Sex Determination Genes. Expression levels of *Csd*, *Fem*, and *Tk* transcripts across drone, queen, and worker honeybees (*n* = 6 per caste). Box plots show median, interquartile range, and mean ± SE (red diamonds with error bars). Statistical significance was determined while employing ANOVA with Bonferroni correction (*** *p* < 0.001).

## Data Availability

Raw data have been deposited to National Center for Biotechnology Information (NCBI) under the BioProject numbers SRS1249139, SRS1263242, SRS1263244, SRS1263211, SRS1263243 and SRS1263256.

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
