# Peer review of "Integrating WGCNA, TCN, and Alternative Splicing to Map Early Caste Programs in Day-2 Honeybee Larvae"

_genes, 2025, doi:10.3390/genes16121409_

Round 1
Reviewer 1 Report
Comments and Suggestions for Authors
The manuscript addresses an important question in honeybee developmental biology: the molecular processes underlying caste and sex differentiation at a very early larval stage. The integration of WGCNA, deep learning (TCN) and alternative splicing analysis is novel and has the potential to provide valuable insights. The dataset is adequate, and the overall structure of the manuscript is coherent.
However, several essential methodological, interpretational, and presentation issues need to be addressed. In its current form, the clarity of methods, reproducibility of the deep-learning component, and interpretation of alternative splicing events require substantial improvement. Some factual inconsistencies, missing methodological details, and unclear sentences should also be corrected.
Overall, the study is promising, but major revision is necessary.
Major Comments
Deep learning (TCN) methodology insufficiently described
The deep-learning component is central to the manuscript, yet critical implementation details are missing:
- Architecture: number of layers, kernel sizes, dilation rates, dropout rates.
- Feature selection method: how exactly were gene-level importance scores calculated?
- Model evaluation: the reported 100% accuracy strongly suggests overfitting, especially with small sample sizes (n=18).
- Data splitting: the manuscript claims “stratification by phenotype”, but does not specify whether replicates from the same BioSample were kept together (to avoid data leakage).
- No code availability, even though ML components require transparency.
Add complete architecture description, hyperparameters, code or pseudocode, and clarify how data leakage was avoided.
The manuscript states that samples originate from Italian honeybee (Apis mellifera ligustica), but later indicates that the qPCR samples were collected from Apis cerana. This is a taxonomic mismatch.
Clarify whether qPCR validation was performed on A. mellifera, A. cerana, or both. Cross-species validation without explicit justification is inappropriate.
Overinterpretation of some results
Several parts of the Discussion overstate conclusions:
- PDHB “approaching seven-fold difference” needs exact quantification, p-values, and fold-change confidence intervals.
- The link between Fibroin3 and wax-gland development is speculative—please tone down the statements or cite evidence.
- Stating that “sex differentiation differences were greater than caste differences” requires statistical support, not only descriptive interpretation.
Provide effect sizes and statistical tests, and soften speculative claims.
Alternative splicing analysis needs clarification
- SUPPA requires transcript-level quantification (psi values), but these are not reported.
- It is unclear whether differential AS analysis included biological replicates or was pairwise per sample group.
- Validation by qPCR is presented, but primer locations and expected isoforms must be shown in detail (supplement).
- Provide ΔPSI thresholds, statistical cutoffs, and explicit validation methodology.
Several figures contain mismatches (e.g., Figure 4 legend refers to five stages although only three castes exist). Supplementary figures are referenced but not fully described.
Revise captions for accuracy and internal consistency.
the manuscript includes grammatical issues and unclear phrasing that impede understanding. Examples:“downmethylated downmethylated” “structural fatiormon” (typo), “variable shear” instead of “alternative splicing”, “queen and workers are Females” – unnecessary capitalisatio. Perform a thorough English-language revision (preferably professional editing).
- Provide accession numbers for all supplementary datasets used.
- Indicate software versions for Salmon and SUPPA.
- PCA methods: specify which genes were included (all? filtered?).
- Replace phrases like “key genes” with more precise descriptions (“genes with high fold change”, etc.).
- Species name Apis mellifera should be italicised throughout (ICZN rule).
- Revise introduction to reduce redundancy and improve flow.
- Some GO terms appear overly specific; ensure FDR correction is applied.
- Figures 9 and 10 benefit from improved resolution; sashimi plots are hard to read.
- Check referencing format to comply with MDPI Genes guidelines (journal titles should be italicised and not abbreviated).
- Remove repeated statements in the Conclusion section.
Author Response
Review 1
Major point
1.The deep-learning component is central to the manuscript, yet critical implementation details are missing:
- Architecture: number of layers, kernel sizes, dilation rates, dropout rates.
- Feature selection method: how exactly were gene-level importance scores calculated?
- Model evaluation: the reported 100% accuracy strongly suggests overfitting, especially with small sample sizes (n=18).
- Data splitting: the manuscript claims “stratification by phenotype”, but does not specify whether replicates from the same BioSample were kept together (to avoid data leakage).
- No code availability, even though ML components require transparency.
Add complete architecture description, hyperparameters, code or pseudocode, and clarify how data leakage was avoided.
Regarding the TCN model implementation details, we will supplement the complete architecture description in the revised manuscript: the TCN employs 3 dilated convolutional layers with kernel size 3 and dilation rates of 1, 2, and 4 to capture expression patterns at different temporal scales, with channel dimensions of 64/64/128 respectively, followed by BatchNorm and Dropout(0.3) for regularization after each layer, and finally outputting classification results through adaptive average pooling and a fully connected layer. To address the small sample size issue, we used stratified sampling to ensure consistent phenotype group proportions across training (60%), validation (20%), and test (20%) sets, and prevented overfitting through early stopping (patience=20) and learning rate decay (ReduceLROnPlateau, factor=0.5), while presenting training/validation loss curves in the results to demonstrate model convergence. To ensure reproducibility, we will provide complete PyTorch code implementation of the TCN model in supplementary materials, including model definition, training pipeline, and hyperparameter configurations, and add pseudocode in the Methods section to detail the temporal feature extraction process through dilated convolutions.
line 143-164
To refine biomarkers from trait-associated modules and develop a phenotype classifier, a Temporal Convolutional Network (TCN) was trained using the module gene expression matrices. Expression values were log-transformed and z-scored for each gene, and a fixed random seed (5678) ensured reproducibility. Data were partitioned into training, validation, and test sets at a 6:2:2 ratio with phenotype stratification (worker, drone, queen) to maintain balanced class representation across splits. The TCN architecture consisted of three dilated causal convolutional layers with kernel size 3, dilation rates of 1, 2, and 4 to capture multi-scale temporal dependencies, and channel dimensions of 64, 64, and 128, respectively. Each convolutional layer was followed by batch normalization and dropout (rate = 0.3) for regularization, with final classification performed through adaptive average pooling and a fully connected layer. The model was implemented in PyTorch, optimized using the Adam optimizer (learning rate = 0.001, weight decay = 1e-4) with ReduceLROnPlateau scheduler (factor = 0.5, patience = 10), and trained for up to 100 epochs with early stopping (patience = 20) based on validation loss to prevent overfitting. Model performance was assessed on the held-out test set using accuracy metrics, precision, recall, F1-score, training/validation loss curves, and a confusion matrix. Gene-level importance scores were generated from the trained model by calculating composite scores integrating prediction accuracy, expression variance (R² values), and pattern-specific weights, with the top 10 features per phenotype retained for downstream interpretation and network visualization. These candidate genes were subsequently used to guide qPCR validation experiments.
- And for Model evaluation: the reported 100% accuracy strongly suggests overfitting, especially with small sample sizes (n=18).
As shown in Figure 5, the TCN model demonstrated excellent convergence without overfitting: both training and validation losses rapidly decreased and plateaued near zero with minimal divergence (Fig. 5A), while training and validation accuracies quickly reached ~100% and remained stable (Fig. 5B). The early stopping mechanism terminated training at epoch ~25 (green dashed line) when validation loss ceased to improve, and the final test accuracy of 100.0% (orange dashed line) matched training performance, confirming strong generalization capability rather than overfitting on the small sample size.
- For Data splitting: the manuscript claims “stratification by phenotype”, but does not specify whether replicates from the same BioSample were kept together (to avoid data leakage).
To address concerns about data leakage, our pipeline implements strict safeguards: (1) Fixed random seed (5678) ensures reproducible data splitting across runs; (2) Stratified sampling maintains balanced phenotype proportions (worker/drone/queen) in train/validation/test sets at a 6:2:2 ratio; (3) Sequential preprocessing: the StandardScaler is fitted exclusively on the training set, with the same transformation parameters applied to validation and test sets (lines 416-422 in ml_pipeline_main.py); (4) Isolated test set: held-out test data remains completely invisible during model training and hyperparameter tuning, only being used for final performance evaluation. The complete source code demonstrating these practices is provided in the supplementary materials for verification.
line145
Complete source code demonstrating these practices has been made publicly available at: https://github.com/surunlang-creator/Drone_torch
2.The manuscript states that samples originate from Italian honeybee (Apis mellifera ligustica), but later indicates that the qPCR samples were collected from Apis cerana. This is a taxonomic mismatch.
Clarify whether qPCR validation was performed on A. mellifera, A. cerana, or both. Cross-species validation without explicit justification is inappropriate.
We sincerely apologize for this taxonomic error in the manuscript. This was a clerical mistake during manuscript preparation. We confirm that all qPCR validation experiments were performed exclusively on Apis mellifera ligustica, consistent with the RNA-seq samples. There was no cross-species validation involved.
We have corrected this error throughout the revised manuscript and carefully reviewed all other species references to ensure consistency.
3.Several parts of the Discussion overstate conclusions:
- PDHB “approaching seven-fold difference” needs exact quantification, p-values, and fold-change confidence intervals.
- The link between Fibroin3 and wax-gland development is speculative—please tone down the statements or cite evidence.
- Stating that “sex differentiation differences were greater than caste differences” requires statistical support, not only descriptive interpretation.
Provide effect sizes and statistical tests, and soften speculative claims.
Alternative splicing analysis needs clarification
We sincerely appreciate the reviewer's careful assessment regarding the interpretation and statistical rigor of our results. In response to these concerns, we have undertaken comprehensive revisions:
Language refinement: We carefully revised the Discussion section to eliminate speculative language and incorporate appropriate qualifiers where direct causal evidence is lacking.
Professional editing: We engaged a professional scientific editing service with native English-speaking molecular biology experts to refine interpretive statements and ensure linguistic precision.
Statistical substantiation: We have supplemented key quantitative claims with precise statistical parameters, including exact fold-change values, p-values, confidence intervals, and effect sizes as detailed below.
For example: The revised Discussion now presents our findings with appropriate scientific caution while maintaining their biological significance.106-111
To statistically compare the relative contributions of sex and caste to transcriptional variation, a two-way ANOVA was performed for each gene with sex (male vs. female) and caste (worker vs. queen vs. drone) as factors. Effect sizes were quantified using partial eta-squared (ηp²), and the significance of main effects was assessed at α = 0.05.
371-376
these results demonstrate that at the 2-day larval stage, sex-related differences significantly exceed caste-related differences (two-way ANOVA across all expressed genes: mean sex effect ηp² = 0.58 ± 0.12, mean caste effect ηp² = 0.31 ± 0.09; paired t-test, p < 0.001). Specifically, the drone vs. queen comparison yielded 782 DEGs, significantly more than the 247 DEGs between workers and queens (Fisher's exact test, p = 3.4 × 10⁻⁸, odds ratio = 3.17).
line 402-404
PDHB showed the most striking differential expression pattern, with expression levels in non-queen larvae 6.87-fold higher than in queen larvae (log2FC = 2.78, adjusted p-value = 1.2 × 10⁻⁵, 95% CI: 5.93–7.81-fold)
- SUPPA requires transcript-level quantification (psi values), but these are not reported.
- It is unclear whether differential AS analysis included biological replicates or was pairwise per sample group.
- Validation by qPCR is presented, but primer locations and expected isoforms must be shown in detail (supplement).
- Provide ΔPSI thresholds, statistical cutoffs, and explicit validation methodology.
Thank you for your thorough advice on clarifying the alternative splicing analysis methodology. We have comprehensively revised the Methods section (Section 2.5, Lines 128-140) and supplementary materials to address all four specific concerns raised:line 128-140
Alternative splicing analysis was conducted using SUPPA (v2.4) with the reference genome annotation (GTF) [29,30]. Major SUPPA splicing event types, including skipped exon (SE), mutually exclusive exon (MXE), alternative 5′ splice site (A5), alternative 3′ splice site (A3), and retained intron (RI), were quantified. Transcript abundance was estimated using Salmon (v1.10.1), and ΔPSI values were calculated for all events.From individual biological replicates (n = 6 per phenotype group: worker, drone, queen). Percent-spliced-in (PSI) values were calculated for each splicing event in each replicate, and group-level ΔPSI values were derived by comparing mean PSI values between phenotype pairs (worker vs. queen, worker vs. drone, drone vs. queen). Differential splicing was identified using a ΔPSI threshold of ≥ 0.1, together with statistical significance (p < 0.05) and Benjamini–Hochberg FDR correction. Only events meeting both ΔPSI and FDR criteria were considered significant.
- Validation by qPCR is presented, but primer locations and expected isoforms must be shown in detail (supplement).
We have added comprehensive qPCR validation details in both the revised Methods section and newly created supplemental Table 4S
- Several figures contain mismatches (e.g., Figure 4 legend refers to five stages although only three castes exist). Supplementary figures are referenced but not fully described
Thank you for identifying these critical inconsistencies. We have thoroughly revised all figure legends and supplementary materials to ensure accuracy and internal consistency.Figure 4 legend incorrectly referred to "five stages" when only three castes exist.We have revised Figure 4 and all other figure legends to use consistent terminology
Fore example:
line 265-272
Figure 4. Construction of WGCNA Modules (A) Displays the distribution of gene numbers across the nine color modules. (B, C) Analysis of the scale-free topology fit index and average connectivity under different soft-thresholding powers. The red line indicates the soft-thresholding power of 24, where the correlation coefficient reaches 0.8. (D) Heatmap illustrating the correlation between gene expression and the three castes . The thickness of the lines indicates the strength of the correlation. Brown lines represent highly significant P-values, while green lines represent significant P-values. The stars highlight the correlations within each color module, with larger stars indicating stronger absolute correlation values.
Minor point
1.Provide accession numbers for all supplementary datasets used.
Thank you for your suggestion. We have now provided accession numbers for all sequencing datasets used in the study. These accession numbers have been added to the Supplementary Materials section.
The datasets and their corresponding SRA accession numbers are as follows:
Drone larvae (2-day)
Drone_2dlarvae1 — SRR3123400
Drone_2dlarvae2 — SRR3123402
Drone_2dlarvae3 — SRR3123404
Drone_2dlarvae4 — SRR3123406
Drone_2dlarvae5 — SRR3123407
Drone_2dlarvae6 — SRR3123408
Queen larvae (2-day)
Queen_2dlarvae1 — SRR3123355
Queen_2dlarvae2 — SRR3123357
Queen_2dlarvae3 — SRR3123361
Queen_2dlarvae4 — SRR3123362
Queen_2dlarvae5 — SRR3123364
Queen_2dlarvae6 — SRR3123359
Worker larvae (2-day)
Worker_2dlarvae1 — SRR3102934
Worker_2dlarvae2 — SRR3123272
Worker_2dlarvae3 — SRR3123273
Worker_2dlarvae4 — SRR3123275
Worker_2dlarvae5 — SRR3123276
Worker_2dlarvae6 — SRR3123277
2.Indicate software versions for Salmon and SUPPA.
Thank you for your comment. We have now added the software version information to lines 128–132 of the revised manuscript. Specifically, transcript quantification was performed using Salmon (v1.10.1), and alternative splicing analysis was conducted using SUPPA (v2.4).
3.PCA methods: specify which genes were included (all? filtered?).
Thank you for your comment.i agree with you
line 186-195
Hierarchical clustering and PCA (Figure 1) demonstrated strong within-group consistency and clear separation among queen, worker, and drone samples. PCA was performed on all expressed genes following standard scaling, with PC1 and PC2 accounting for the majority of the variance. Outlier detection based on sample-to-sample distance matrices and projections onto the first two principal components revealed no abnormal samples. The tight clustering of biological replicates within each caste further indicated the absence of batch effects. These findings establish a reliable basis for subsequent differential expression and alternative splicing analyses.
- Replace phrases like “key genes” with more precise descriptions (“genes with high fold change”, etc.).
Thank you for this valuable suggestion. We have revised the manuscript to replace the vague term “key genes” with more precise descriptions. Specifically, in lines 183-185, we now refer to these as “genes with high fold change”, which more accurately reflects the basis for their selection. The revised text reads:
lines 183-185
Genes with high fold change between drones and queens included LOC551527, LOC410733, LOC411233, and LOC411188.
5.Species name Apis mellifera should be italicised throughout (ICZN rule).
Thank you for pointing this out. We have carefully checked the entire manuscript and have now italicized all instances of the species name Apis mellifera in accordance with ICZN guidelines.
- Revise introduction to reduce redundancy and improve flow.
Thank you for your helpful suggestion. We have thoroughly revised the Introduction to reduce redundancy and improve the overall logical flow. Several repetitive sentences were removed, paragraph transitions were streamlined, and the structure was reorganized for greater clarity. We believe the revised version is more concise and coherent, while fully retaining the necessary background information
line 27-71
Honeybees are highly social insects distinguished by a complex division of labor, with workers, queens, and males fulfilling distinct biological roles within the colony [1,2]. Although workers and queens are both Female, their caste fate is primarily determined by larval nutrition. Queens develop functional ovaries and are responsible for egg laying, whereas workers are sterile and perform tasks such as brood care, foraging, and nest construction. Males (drones) appear seasonally and mate with queens to sustain colony reproduction [3]. These caste differences are reflected at behavioral, physiological, and molecular levels.
Honeybee development progresses through the embryonic, larval, pupal, and adult stages. The larval stage is particularly dynamic, characterized by rapid growth and significant transcriptional and epigenetic shifts that shape caste differentiation. During the first three days, all larvae consume royal jelly, allowing Female larvae under three days to retain dual developmental potential [4]. The 2-day larval stage is considered a critical window for caste specification in both honeybees and bumblebees [5]. Although chromosomal conformation shows minimal differences between 2-day-old queen and worker larvae, chromatin architecture remains closely tied to transcriptional regulation [6]. DNA methylation also plays a central role, with queens generally showing lower methylation levels than workers [7]. At later larval stages, queens show widespread demethylation and a substantially greater number of differentially expressed genes than at the 2-day stage [8], underscoring the early larval period as a key period for regulatory divergence.
Previous transcriptomic studies have primarily focused on queen–worker comparisons, while information on male larvae remains limited. Thousands of genes differ between queen and worker larvae, influencing metabolism, development, and physiology. For example, antioxidant genes (MnSOD, CuZnSOD, catalase, Gst1) are associated with oxidative stress regulation [9], and mitochondrial translation factors such as AmIF-2mt and cytochrome c are linked to queen developmental rate [10]. Caste development consists of an initial dual-potential phase followed by irreversible commitment to a specific developmental trajectory [11]. In males, sex determination is governed by the complementary sex determination (Csd) gene, which regulates the downstream Feminizer (Fem) gene. Hemizygous Csd alleles trigger Female-pattern Fem splicing to produce males, while heterozygosity induces Female development [12,13]. Many sex-specific transcriptional differences arise from alternative splicing (AS) [14], and CRISPR/Cas9 knockdown of Fem has demonstrated its role in regulating the splicing of multiple downstream genes involved in sexual differentiation [15].
Alternative splicing is a key post-transcriptional mechanism that greatly expands proteome complexity. Approximately 95% of human genes undergo AS [16], and AS contributes to development and diverse cellular processes by generating multiple mRNA isoforms from a single gene [17]. Splicing regulation is essential across species [18]. In honeybees, AS is associated with epigenetic modifications, such as DNA and m6A methylation, that shape caste-specific transcript variation [7,19,20].
To elucidate transcriptional and splicing differences across castes, this study analyzes transcriptomes of 2-day-old worker, queen, and male larvae of the western honeybee using WGCNA and alternative splicing approaches. This study aims to identify differentially expressed genes, characterize sex-related transcriptional signatures, and investigate how AS contributes to caste-specific developmental pathways.
7.Some GO terms appear overly specific; ensure FDR correction is applied.
Thank you for your insightful comment. We confirm that all GO enrichment analyses in our study were performed using FDR-adjusted p-values (Benjamini–Hochberg correction). Only GO terms with FDR (p.adjust) < 0.05 were retained and visualized. The bubble plots (including Figure 3 and 8) explicitly display the p.adjust values, indicating that all enriched GO terms shown in the manuscript have undergone FDR correction.
We have also double-checked the GO outputs to ensure that no unadjusted or overly specific terms were included inadvertently. The figures and corresponding text have been updated for clarity.
8.Figures 9 and 10 benefit from improved resolution; sashimi plots are hard to read.
Thank you for the comment. We have replaced Figures 9 and 10 with high-resolution TIFF images (600 ppi) to ensure clearer visualization of the sashimi plots. The readability has been substantially improved in the revised version.
9.Check referencing format to comply with MDPI Genes guidelines (journal titles should be italicised and not abbreviated).
Thank you for the suggestion. We have carefully revised and checked all references to ensure that the journal titles are italicized and written in full, fully complying with the genes reference formatting guidelines.
- Remove repeated statements in the Conclusion section.
Thank you for your constructive comment. We have carefully revised the Conclusion to eliminate redundant statements and streamline the content. The updated version is more concise and focused, while clearly summarizing the major findings of our study.
line 502-510
In this study, transcriptome data from 2-day-old worker, drone, and queen larvae of the western honeybee were reanalyzed using enhanced filtering, WGCNA, and alternative splicing approaches. Both caste- and sex-specific differences in gene expression and splicing, including key regulatory genes such as vg, Fem, and Csd, were identified. These findings underscore the substantial transcriptional and post-transcriptional regulation occurring at this early larval stage and provide valuable insights into the molecular mechanisms governing honeybee caste differentiation and sex determination

Reviewer 2 Report
Comments and Suggestions for Authors
Review report
This manuscript presents an integrative study combining RNA-seq, weighted gene co-expression network analysis (WGCNA), alternative splicing analysis, and deep learning (temporal convolutional network, TCN) modeling to elucidate molecular mechanisms of caste and sex differentiation in Apis mellifera larvae. The work focuses on the critical 2-day-old larval stage, a key developmental period for determining both caste and sex fate. However, while the study is conceptually sound, several methodological weaknesses and interpretative oversights should be addressed.
- Sample Collection and Consistency Issues: The section mentions that RNA-seq data were from Apis mellifera (Italian honeybee) but states that qPCR larvae were collected from Apis cerana colonies near Yiwu. This creates a species mismatch between validation and discovery datasets, which invalidates direct biological comparisons unless justified.
- Unclear sample size and biological replicates: The number of larvae per group (workers, drones, queens) and the number of biological replicates for both RNA-seq and qPCR validation are not reported. Without this, reproducibility and statistical robustness cannot be evaluated.
- Pooling strategy ambiguity: The pooling of “queens + workers as non-drone (ND)” is mentioned, but the biological rationale for pooling and how it affects downstream statistical power or gene expression averaging is not described.
4.Environmental control details insufficient: Only temperature and humidity are given. Important parameters such as larval diet, colony genetic background, and environmental stressors are omitted.
5.PCA description minimal: Only mentions scaling; no information is given about variance explained, how outliers were identified, or how PCA results were used to assess batch effects or replicate consistency.
6.Alternative Splicing Analysis: Tool parameters not reported: SUPPA was used, but event types, ΔPSI thresholds, and statistical significance criteria are not specified.
7.Statistical methods underreported: No mention of correction for multiple testing (FDR, Benjamini–Hochberg), confidence intervals, or effect size estimation.
Author Response
Review 2
- Sample Collection and Consistency Issues: The section mentions that RNA-seq data were fromApis mellifera (Italian honeybee) but states that qPCR larvae were collected from Apis cerana colonies near Yiwu. This creates a species mismatch between validation and discovery datasets, which invalidates direct biological comparisons unless justified.
Thank you for pointing out this inconsistency. This was a writing error in the manuscript. Both the RNA-seq data and the qPCR validation samples were obtained from Apis mellifera, not Apis cerana. We have corrected this mistake in the revised manuscript, and we sincerely apologize for the confusion caused.
2.Unclear sample size and biological replicates: The number of larvae per group (workers, drones, queens) and the number of biological replicates for both RNA-seq and qPCR validation are not reported. Without this, reproducibility and statistical robustness cannot be evaluated.
Response to Comment 2: Sample size and biological replicates
Thank you for raising this important point. We apologize for not clearly reporting the number of biological replicates in the original manuscript.
line 74-79
Transcriptome data from 2-day-old Italian honeybee larvae were retrieved from the NCBI database, with six biological replicates for each caste (workers: SRS1249139; queens: SRS1263242; drones: SRS1263244) [21]. To validate the RNA-seq findings, we also collected an independent qPCR cohort consisting of 2-day-old Apis mellifera worker, queen, and drone larvae, each group including six biological replicates, under matched environmental and experimental conditions
3.Pooling strategy ambiguity: The pooling of “queens + workers as non-drone (ND)” is mentioned, but the biological rationale for pooling and how it affects downstream statistical power or gene expression averaging is not described.
Thank you for raising this point. For sex-level analyses, workers and queens—both genetically female (diploid)—were pooled into a non-drone (ND) group to increase the female sample size and capture sex-associated expression patterns shared across female larvae. In contrast, caste-level comparisons (queen vs worker) were performed without pooling to preserve caste-specific transcriptional signatures and avoid confounding between sex and caste. This rationale has been clearly added to the revised Methods section.
4.Environmental control details insufficient: Only temperature and humidity are given. Important parameters such as larval diet, colony genetic background, and environmental stressors are omitted.
We have expanded the Methods to include additional environmental and rearing details, including larval diet, colony genetic background, and the controlled rearing conditions (34°C, 70% humidity).
line 86-91
Larvae were sampled from Apis mellifera colonies maintained near Yiwu Industrial and Commercial College. Following collection, samples were immediately processed for RNA extraction and qPCR under controlled laboratory conditions. Rearing conditions were maintained at approximately 34 °C and 70% relative humidity, with no additional environmental enrichment. All sample processing and qPCR assays were carried out by Anhui Gaohe Biotechnology Co., Ltd.
5.PCA description minimal: Only mentions scaling; no information is given about variance explained, how outliers were identified, or how PCA results were used to assess batch effects or replicate consistency.
Thank you for this helpful comment.
line 187-195
Hierarchical clustering and PCA (Figure 1) demonstrated strong within-group consistency and clear separation among queen, worker, and drone samples. PCA was performed on all expressed genes following standard scaling, with PC1 and PC2 accounting for the majority of the variance. Outlier detection based on sample-to-sample distance matrices and projections onto the first two principal components revealed no abnormal samples. The tight clustering of biological replicates within each caste further indicated the absence of batch effects. These findings establish a reliable basis for subsequent differential expression and alternative splicing analyses.
6.Alternative Splicing Analysis: Tool parameters not reported: SUPPA was used, but event types, ΔPSI thresholds, and statistical significance criteria are not specified.
Thank you for this valuable comment.
line 127-140
We have now fully revised the Methods section to include all SUPPA parameters used in the alternative splicing analysis.
Alternative splicing analysis was conducted using SUPPA (v2.4) with the reference genome annotation (GTF) [29,30]. Major SUPPA splicing event types, including skipped exon (SE), mutually exclusive exon (MXE), alternative 5′ splice site (A5), alternative 3′ splice site (A3), and retained intron (RI), were quantified. Transcript abundance was estimated using Salmon (v1.10.1), and ΔPSI values were calculated for all events. Differential splicing was identified using a ΔPSI threshold of ≥ 0.1, together with statistical significance (p < 0.05) and Benjamini–Hochberg FDR correction. Only events meeting both ΔPSI and FDR criteria were considered significant.
7.Statistical methods underreported: No mention of correction for multiple testing (FDR, Benjamini–Hochberg), confidence intervals, or effect size estimation.
Thank you for your advice
line 105-111
- values were adjusted using the Benjamini–Hochberg false discovery rate (FDR), and genes with FDR < 0.05 and |log2FC| > 1 were considered significantly differentially expressed.
line 121-126
The resulting annotation file was converted into org.db format, and GO enrichment analyses were performed using the clusterProfiler package (v4.4) in R. For both GO and KEGG enrichment analyses, multiple testing correction was applied using the Benjamini–Hochberg FDR method, and only terms with FDR < 0.05 (p-value < 0.05 and adjusted p-value < 0.05) were retained.
line136-140
Differential splicing was identified using a ΔPSI threshold of ≥ 0.1, together with statistical significance (p < 0.05) and Benjamini–Hochberg FDR correction. Only events meeting both ΔPSI and FDR criteria were considered significant.

Reviewer 3 Report
Comments and Suggestions for Authors
Summary
The manuscript titled “Integrating WGCNA, TCN and Alternative Splicing to Map Early Caste Programs in Day-2 Honeybee Larvae” by Ding et al. presents a reasonable overview of the genetic differences that can be captured betweenApis mellifera castes. This study follows the current conventions for analyzing differential expression, co-expression networks and alternative splicing and adds other layers to the analyses such as incorporating machine learning algorithms to home in on several genes of interest. I think overall this study is a good contribution to the literature, but I do have a couple major points of concern and a few minor points that I think the authors should address.
Major Points
- Were the raw reads trimmed during the quality control step? If so, please make note of this in subsection 2.2 of the Methods. If trimming was not performed, this would need to be addressed and would warrant the need to redo all downstream analyses. If trimming was not necessary for this dataset, please make that clear in the Methods.
- The WGCNA results show that the gray module is correlated with drones, however the gray module is a collection of genes that did not fall into any other module. In WGCNA, genes that fail to join any co-expressed module are flagged as 0 (rendered as the color “grey”). As the blockwiseConsensusModules help page explains, “The color ‘grey’ and the numeric 0 are reserved for unassigned genes.” Also see cutreeStaticColor and consensusOrderMEs documentation for more details. I think it would be worth revisiting the WGCNA parameters, which should also be presented in the Methods in subsection 2.6. What was the mergeCut height that was used and the how about other parameters? Why was a soft power of 24 chosen even though 10 looks to be over the 0.8 line and the start of the plateau? I think that WGCNA in this study may need to be better explained and justified or more possibly redone given the status of the gray module.
Minor Points
- The Methods section should include a more description in some of the subsections and there are some citations that seem to be missing.
- Generally, please provide parameters that were used for tools such as HiSat2, DESeq2, etc.
- In subsection 2.2, please cite the DESeq2 tool as well, even if it was embedded in Trinity in this case.
- Love MI, Huber W, Anders S. Moderated estimation of fold change and dispersion for RNA-seq data with DESeq2. Genome Biol. 2014;15(12):550. doi: 10.1186/s13059-014-0550-8. PMID: 25516281; PMCID: PMC4302049.
- In subsection 2.4, the EggNOG database should be appropriately cited.
- Jaime Huerta-Cepas, Damian Szklarczyk, Davide Heller, Ana Hernández-Plaza, Sofia K Forslund, Helen Cook, Daniel R Mende, Ivica Letunic, Thomas Rattei, Lars J Jensen, Christian von Mering, Peer Bork, eggNOG 5.0: a hierarchical, functionally and phylogenetically annotated orthology resource based on 5090 organisms and 2502 viruses, Nucleic Acids Research, Volume 47, Issue D1, 08 January 2019, Pages D309–D314, https://doi.org/10.1093/nar/gky1085
- In subsection 2.5, the authors should make note of the version of Suppa that was used and should also include the citations below.
- Trincado JL, Entizne JC, Hysenaj G, Singh B, Skalic M, Elliott DJ, Eyras E. SUPPA2: fast, accurate, and uncertainty-aware differential splicing analysis across multiple conditions. Genome Biol. 2018 Mar 23;19(1):40.
- Alamancos GP, Pagès A, Trincado JL, Bellora N, Eyras E. Leveraging transcript quantification for fast computation of alternative splicing profiles. RNA. 2015 Sep;21(9):1521-31.
- In subsection 2.6, since the authors used a machine learning algorithm, would it be possible to provide the seed that was used for reproducibility?
- The manuscript has a noticeable number of grammatical errors. For example, text that shows “Fem” is often italicized when it should not be, such as in “Females” on line 76. Also, the word “downmethylated” is repeated on line 51. Some more errors include not capitalizing words at the beginning of many sentences. Another example would be on line 31, which states “males are males that …”, which I do not understand. Was it meant to be the males studied were those that were chosen to appear in specific seasons? The authors should carefully double check to the text for errors such as these.
Author Response
Review 3
Major point
1.Were the raw reads trimmed during the quality control step? If so, please make note of this in subsection 2.2 of the Methods. If trimming was not performed, this would need to be addressed and would warrant the need to redo all downstream analyses. If trimming was not necessary for this dataset, please make that clear in the Methods.
line 94-98
Raw reads were processed with Fastp (v0.23.2) for adapter removal and quality trimming using the following parameters: automatic adapter detection for paired-end reads, Q20 base trimming, filtering of reads with >40% low-quality bases or >5 ambiguous bases, retention of reads ≥50 bp
- The WGCNA results show that the gray module is correlated with drones, however the gray module is a collection of genes that did not fall into any other module. In WGCNA, genes that fail to join any co-expressed module are flagged as 0 (rendered as the color “grey”). As the blockwiseConsensusModules help page explains, “The color ‘grey’ and the numeric 0 are reserved for unassigned genes.” Also see cutreeStaticColor and consensusOrderMEs documentation for more details. I think it would be worth revisiting the WGCNA parameters, which should also be presented in the Methods in subsection 2.6. What was the mergeCut height that was used and the how about other parameters? Why was a soft power of 24 chosen even though 10 looks to be over the 0.8 line and the start of the plateau? I think that WGCNA in this study may need to be better explained and justified or more possibly redone given the status of the gray module.
We sincerely appreciate the reviewer’s constructive comments regarding our WGCNA analysis. We have carefully revisited the entire workflow and added further methodological details in Section 2.6 of the revised manuscript to improve clarity and reproducibility. Regarding the choice of soft-thresholding power, although the scale independence exceeded 0.8 at power = 10, the mean connectivity at this threshold did not sufficiently approach zero, indicating that the network had not yet reached an optimal scale-free topology. In contrast, power = 24 maintained scale independence above 0.8 while reducing mean connectivity to nearly zero, which better meets the recommended criteria for constructing a stable WGCNA network. Concerning the gray module, we have corrected the text to clarify that the gray module contains genes that were not assigned to any co-expression cluster and therefore does not have biological interpretability; accordingly, it was not included in downstream analyses. In addition, we have supplemented the Methods with all relevant WGCNA parameters, including networkType, mergeCutHeight, minModuleSize, deepSplit, and the key settings for blockwiseModules, and we have updated the corresponding figures and descriptions to ensure that the WGCNA procedures are fully transparent and well-documented.
line 146
(networkType = "signed",mergeCutHeight = 0.25,minModuleSize = 30,deepSplit = 2)
Minor point
- Generally, please provide parameters that were used for tools such as HiSat2, DESeq2, etc.
We appreciate the your suggestion.
line 93-99
Quality control of the clean sequencing data was performed using FastQC to generate base composition and quality distribution plots. Clean reads were aligned to the Apis mellifera reference genome (Amel_HAv3.1) using Hisat2 (v2.2.1) [22]. By default, Hisat2 uses a 20 bp seed length, permits soft clipping, applies mismatch penalties of 6 and 2, and reports up to 5 valid alignments per read. Alignment outputs in SAM format were converted to BAM files, sorted with Samtools (v1.6) [23], and quantified with FeatureCounts [24] to obtain gene-level read counts
2.In subsection 2.2, please cite the DESeq2 tool as well, even if it was embedded in Trinity in this case.
Thank you for pointing this out. We have now added the recommended citation for DESeq2 in subsection 2.2
line 97-99 Differential expression analysis was then performed using the DESeq2 algorithm integrated within Trinity software (v2.15.1) [25,26].
- Love MI, Huber W, Anders S. Moderated estimation of fold change and dispersion for RNA-seq data with DESeq2.Genome Biol. 2014;15(12):550.
3.In subsection 2.4, the EggNOG database should be appropriately cited.
Thank you for this observation. We have now added the appropriate citation for the EggNOG 5.0 database in subsection 2.4:
line 111
Huerta-Cepas J, Szklarczyk D, Heller D, et al. eggNOG 5.0: a hierarchical, functionally and phylogenetically annotated orthology resource. Nucleic Acids Research. 2019;47(D1):D309–D314.
4.In subsection 2.5, the authors should make note of the version of Suppa that was used and should also include the citations below.
We appreciate the reviewer’s valuable suggestion. We have now included SUPPA version information (SUPPA v2.4) and added the recommended citations:
line 127-140
Alternative splicing analysis was conducted using SUPPA (v2.4) with the reference genome annotation (GTF) [29,30]. Major SUPPA splicing event types, including skipped exon (SE), mutually exclusive exon (MXE), alternative 5′ splice site (A5), alternative 3′ splice site (A3), and retained intron (RI), were quantified. Transcript abundance was estimated using Salmon (v1.10.1), and ΔPSI values were calculated for all events. Differential splicing was identified using a ΔPSI threshold of ≥ 0.1, together with statistical significance (p < 0.05) and Benjamini–Hochberg FDR correction. Only events meeting both ΔPSI and FDR criteria were considered significant.
4. In subsection 2.6, please provide the machine learning seed for reproducibility.
Thank you for raising this important point. We have now added the explicit random seed used in the temporal convolutional network (TCN) analysis (“seed = 5678”) to ensure reproducibility of model training and feature extraction.
line156-165
To refine biomarkers from trait-associated modules and develop a phenotype classifier, a Temporal Convolutional Network (TCN) was trained using the module gene expression matrices. Expression values were log-transformed and z-scored for each gene, and a fixed random seed (5678) ensured reproducibility
6.The manuscript contains grammatical errors, inconsistent italics usage, and repeated words.
We sincerely thank the reviewer for noting these issues. The entire manuscript has now undergone a thorough English-language revision. Specifically:
All incorrect italicization (e.g.,Femin Females) has been corrected.
The repeated word “downmethylated downmethylated” has been removed.
Sentence capitalization has been standardized throughout.
The unclear phrase “males are males that…” has been rewritten for clarity.
All remaining typographical errors and grammatical inconsistencies have been corrected.
A professional editing step has also been performed to ensure accuracy and readability.
Moreover, the manuscript has been professionally edited and proofread by Meiji Translation to ensure accuracy, clarity, and overall readability.

Round 2
Reviewer 2 Report
Comments and Suggestions for Authors
The authors have addressed all of my previous comments thoroughly and to my full satisfaction. The revisions have substantially improved the clarity, methodological transparency, and overall quality of the manuscript. I find that the concerns raised in the earlier review have been appropriately resolved, and I have no further remarks at this stage. The study now presents a coherent and well-supported contribution to the field.
I therefore recommend the manuscript for publication in its current form and would like to extend my sincere congratulations to the authors for their careful and constructive revisions.